# The Effect of Recycling on Wood-Fiber Thermoplastic Composites

**DOI:** 10.3390/polym12081750

**Published:** 2020-08-05

**Authors:** Luísa Rosenstock Völtz, Irangeli Di Guiseppe, Shiyu Geng, Kristiina Oksman

**Affiliations:** Division of Materials Science, Department of Engineering Sciences and Mathematics, Luleå University of Technology, SE-971-87 Luleå, Sweden; Luisa.voltz@ltu.se (L.R.V.); iradiyu@gmail.com (I.D.G.); shiyu.geng@ltu.se (S.G.)

**Keywords:** wood fiber, thermoplastic composite, extrusion, mechanical properties, recycling

## Abstract

The aim of this study was to investigate the effect of recycling on polypropylene (PP) and wood-fiber thermoplastic composites (WPCs) using a co-rotating twin-screw extruder. After nine extrusion passes microscopy studies confirmed that the fiber length decreased with the increased number of recycling passes but the increased processing time also resulted in excellent dispersion and interfacial adhesion of the wood fibers in the PP matrix. Thermal, rheological, and mechanical properties were studied. The repeated extrusion passes had minimal effect on thermal behavior and the viscosity decreased with an increased number of passes, indicating slight degradation. The recycling processes had an effect on the tensile strength of WPCs while the effect was minor on the PP. However, even after the nine recycling passes the strength of WPC was considerably better (37 MPa) compared to PP (28 MPa). The good degree of property retention after recycling makes this recycling strategy a viable alternative to discarding the materials. Thus, it has been demonstrated that, by following the most commonly used extrusion process, WPCs can be recycled several times and this methodology can be industrially adapted for the manufacturing of recycled products.

## 1. Introduction

Environmental regulations are placing pressure on manufacturers to consider the environmental impact of their products. By replacing conventional materials with renewable and biodegradable materials such as natural fibers, or reusing materials by reprocessing them several times [1,2], environmental impact can be avoided. From a material point of view, the recycling and reuse of materials present an excellent opportunity for producing new products with comparable properties. From an industrial point of view, mechanical recycling by means of extrusion and injection molding is the most appropriate alternative, due to low cost and reliability [3]. From the environmental point of view, the natural fibers are renewable, abundant, and biodegradable [4].

Wood-polymer composites (WPCs) are a group of materials mainly consisting of a thermoplastic polymer as a matrix, wood as reinforcement, and a minor amount of additives such as lubricants and coupling agent [5]. Due to the increase in consumer ecological awareness, the WPCs have gained more attention as an alternative to traditional polymeric materials [6,7]. Recycled plastics have been used for the production of WPCs since the 1990s with an expressed increase in the last few years [8]. However, nationwide WPCs recycling systems are still missing [7]. By understanding the properties of the recycled WPCs, the re-processing of WPCs can be adjusted and the relationship between physical and mechanical properties can be used to establish a new circular-economy approach for sustainability. Nevertheless, several limitations have to be taken into consideration, such as thermal degradation, polymer chain scission resulting in a reduction of molecular weight, and the decrease of the wood fiber length due to the shear histories [7,9]. These changes could also affect the interfacial bonding between the polymeric matrix and wood fibers, resulting in a decrease in the mechanical properties [10].

Dickson et al. [2] presented the effect of six extrusion cycles on wood fiber/polypropylene composites, showing a decrease of 21% in tensile strength and 17% in the tensile modulus.

Shahi et al. [10] investigated the recyclability of composites consisting of wood flour (60 wt.%) and high density polyethylene (HDPE) (40 wt.%). They showed that the mechanical properties of the composites were reduced already after a single recycling step. The reason for the decrease was cited as being caused by the poor adhesion between the HDPE and wood flour.

Le Duigou et al. [11] studied the effect of recycling on flax fiber poly(l-lactide) composites after extrusion with a single screw extruder followed by six injection molding cycles. They reported a 20% decrease in the fiber length/diameter ratio (l/d) after the extrusion which was further decreased by 70% after the injection molding cycles. They also showed that the tensile strength decreased by 70% after the six injection molding cycles.

Beg and Pickering [12] recycled cellulose fiber/polypropylene (PP) composites up to eight times using a twin-screw extruder and reported that tensile strength was reduced by 25% and tensile modulus by 16%, because of the severe reduction in the fiber length [12].

Admitting that the tendency is usually the reduction of mechanical properties with the recycling times, recycling can also bring an improvement in the stiffness, which is attributed to the increased dispersion of the fiber within the polymer matrix [2,12].

In this work, the effect of the repeated nine extrusion processes on the properties of thermomechanical pulp polypropylene composites and neat polypropylene was studied. The material characterization was first performed on the raw material (before recycling) in order to quantify the initial reference properties. After the first, third, fifth, seventh and ninth recycles, the wood fiber size and the l/d, shear viscosity, thermal and mechanical properties, and how the repeated extrusion cycles affected the microstructure of the materials including dispersion and interface were studied.

## 2. Materials and Methods

### 2.1. Raw Materials

Commercial wood fiber thermoplastic composite (WPC) compound, consisting of 40 wt.% thermomechanical pulp (TMP), 60 wt.% polypropylene (PP) and a minor amount of maleic anhydride polypropylene (MAPP) coupling agent was kindly supplied by Stora Enso (Hylte Mill, Hylte, Sweden). The PP is a copolymer polypropylene (C-7069-100NA, Braskem Netherlanads, Rotterdam, Netherlands) with a melt flow index (MFI) of 100 g/min (230 °C/2.16 kg). The 98% purity xylene purchased from VWR International AB (Spånga, Sweden) and ethanol (99.5%, CCS Healthcare AB, Borlänge, Sweden) were used for Soxhlet extraction. Distilled water was used for final washing.

### 2.2. Recycling by Twin-Screw Extrusion

The processes for recycling the WPCs were simulated by passing the WPC pellets through a co-rotating twin-screw extruder (Coperion W&P ZSK-18 MEGALab, Stuttgart, Germany) nine times. Figure 1a,b show the extruder profile and screw-design with different processing sections comprising conveying elements, kneading elements, and mixing elements, respectively. The moisture content of the material prior to the extrusion process was measured using a moisture analyzer, HR83 (Mettler Toledo, OH, USA). The dried WPC (moisture content below 0.2%) was fed using a K-Tron gravimetric feeder (Niederlenz, Switzerland) in the extruder with a 3 kg/h throughput rate. The extruder has seven temperature zones, with the processing temperatures set in 190, 190, 190, 190, 195, 200, and 205 °C, respectively. The same procedure was applied using unfilled PP pellets (moisture below 0.2%). The screw speed for WPC and unfilled PP was 450 and 280 rpm, respectively, due to the difference in the viscosity and torque. The extruder die utilized was a strand die, the extrudate was cooled in a water bath and pelletized using a strand pelletizer. The obtained pellets were dried in an oven at 60 °C overnight. Both materials were recycled in the extruder up to nine times and the material characterizations were performed on the material before recycling (PP-0 and WPC-0), 1st recycling (PP-1 and WPC-1), 3rd recycling (PP-3 and WPC-3), 5th recycling (PP-5 and WPC-5), 7th recycling (PP-7 and WPC-7), and 9th recycling (PP-9 and WPC-9).

### 2.3. Characterization

#### 2.3.1. Fiber Size

The analysis of the fibers was performed to see the effect of the extrusion process on the fiber length and diameter and thus the fiber reinforcing capacity. Since the wood fibers are sensitive to the shear stress, the dimensions of the fibers are expected to be influenced by the recycling process [5,13].

In the present analysis, the fibers were separated from the biocomposites pellets through two steps: (1) Boiling the pellets with xylene (170–175 °C) for 24–48 h, followed by (2) Soxhlet extraction in hot xylene (185 °C) for 6–12 h. Finally, the fibers were filtered and washed with ethanol and distilled water. The extracted fibers were dispersed with water (1.0 wt.%) and analyzed using an optical microscope Eclipse LV100POL (Nikon, Tokyo, Japan). ImageJ NIH software, (National Institutes of Health, Bethesda, MD, USA) was used to measure the length and width of the fibers and, consequently, the aspect ratio. At least 100 fibers were measured for each sample. The one-way analysis of variance (ANOVA) and Tukey’s HSD (Honestly Significantly Different) tests at 5% significance level were applied to the fiber aspect ratio to analyze the influence of the recycling.

#### 2.3.2. Rheology

The viscosity at different shear rates was measured using a capillary rheometer Rheo-tester 1000 (Göttfert, Buchen, Germany) at 230 °C, according to ASTM D-3835. The capillary rheometer has a barrel diameter and length of 12 and 220 mm, respectively. The die is 20 mm long and 1 mm in diameter, with a 0° entry angle. Seven piston speeds in the range of 0.04 to 2.56 mm/s, corresponding to a shear rate range of 50 to 4500 s^−1^ were used to generate the rheological data. It is well known that viscosity is proportional to the molecular weight, which decreases due to the chain scission occurring in the polymer as a consequence of the thermal degradation [14,15], and it can be used as a qualitative measurement for material degradation as a consequence of the recycling process.

#### 2.3.3. Thermal Properties

Polymers and WPC, especially wood fibers that present low thermal stability, can suffer thermal degradation when subject to high temperature during the extrusion process. Therefore, degradation temperature and melting temperature can be related to the decrease in the molecular mass of the polymer as a result of the thermal degradation [16].

The thermal degradation temperature was measured using a Thermo Gravimetric Analyzer TGA Q500 (TA instruments, New Castle, DE, USA) under nitrogen flow. The sample was heated at a heating rate of 10 °C/min within the temperature range of 0–900 °C. To ensure complete polymer melting inside the extruder, the melting temperature was measured by using differential scanning calorimetry—DSC821 (Mettler Toledo, OH, USA). The samples were cooled down to −40 °C and held for 3 min. Next, they were heated until 200 °C at a rate of 10 °C/min with a nitrogen flow of 80 mL/min.

#### 2.3.4. Mechanical Properties

The specimens used for tensile testing were injection molded using a Haake MiniJet Pro Piston Injection Molding System (Thermo Fisher Scientific International, Karlsruhe, Germany). The pellets were subjected to an injection pressure of 400 bar and a cylinder temperature of 205 °C to make dog-bone shaped specimens (type V ASTM D-638).

The mechanical properties were measured using a universal tensile testing machine AGS-X (Shimadzu, Kyoto, Japan) equipped with a 5 kN load cell, a gauge length of 25 mm and a strain rate of 2.5 mm/min, according to ASTM D-638. At least five specimens were tested for each set of materials and the average is presented. The one-way analysis of variance (ANOVA) and Tukey’s HSD (Honestly Significantly Different) tests (significance level of 5%) were applied to the tensile properties with the aim to analyze the effects of the number of extrusion recycling passes on the properties of the materials.

#### 2.3.5. Microstructure

Microstructure, especially the fiber dispersion and interface interaction, in the fracture surfaces after tensile testing was studied using a scanning electron microscope JEOL (JSM-IT300, Tokyo, Japan) using 10 kV acceleration voltage. The fracture surfaces were sputter-coated with platinum (15 nm) to avoid charging.

Fourier transform infrared (FTIR) spectroscopy was performed in order to evaluate the recycling effect in the interaction between wood fibers and PP by the addition of MAPP. A VERTEX 80 (Bruker Optics, Billerica, MA, USA) FTIR spectrophotometer with a range 800–3700 cm^−1^ and a 256−scan resolution was used.

## 3. Results and Discussions

### 3.1. Recycling Effect on Fiber Size

Figure 2 shows the average length of the fibers before and after the recycling process, split in different ranges.

The fibers extracted from the samples before recycling (WPC-0) presented longer fibers, around 2 to 3 mm. As the biocomposites were subjected to the extrusion process, the fiber size decreased, due to the fibrillation process and fiber breakage, as a consequence of the shear and bending forces [13,17]. The fiber shortening started to be more noticeable after the third recycling step (WPC-3). The first two recycles might cause fiber damage, while the complete break only happened during the third recycle. The fiber size decreased considerably after the seventh recycling step (WPC-7). In the WPC-9, more than 70% of the fibers are below 100 µm in length.

Due to the fibrillation process, both length and the diameter were affected and this effect is shown by calculating the l/d (aspect ratio) of the fibers. The aspect ratio of the fibers can be seen in Figure 3 and there is a trend that the aspect ratio is decreased with the increasing number of recycling passes. According to ANOVA and Tukey’s test, the pairs (WPC-0 and WPC-1, WPC-0 and WPC-3, WPC-0 and WPC-5, WPC-0 and WPC-7, WPC-0 and WPC-9, WPC-1 and WPC-7, and WPC-1 and WPC-9) are statistically different, as it can be seen in Appendix A. The decrease in the aspect ratio is due to high shear forces to which the material is subjected inside the extruder, especially in the mixing zones [9], as shown previously in Figure 1.

### 3.2. Rheology Properties

The flow properties of the PP-0, PP-9, WPC-0, WPC-3, WPC-5, and WPC-9 are shown in Figure 4 and a comparison of all PP and WPC materials is shown in Appendix A. For both PP and WPC, the shear rate dependence of the viscosity profile showed a shear-thinning behavior with a decrease in the viscosity as the shear rate increased [18].

In the case of PP, a slight decrease in the viscosity is noticed when comparing PP-0 and PP-9, because the degradation process occurring in the extruder leads to a decrease in molecular weight [17]. Short chains exhibit lower entanglements and better mobility, resulting in lower viscosity [19,20]. The decrease in viscosity for WPCs is expected to be mainly due to the shortening of the fibers, whereas the recycling leads to an increment of the number of fibers particles with shorter length facilitating their alignment in the direction of the flow in the capillary rheometer. Other authors have also reported the same trend [18,21]. Compared to the PP melts, the more significant decrease in viscosity of the WPCs after recycling is not only because of the degradation of the polymeric matrix but also, as described before, due to the decrease of the fiber length.

### 3.3. Thermal Properties

The effect of the recycling passes on thermal properties was evaluated, since the number of extrusion cycles increases the thermal and the mechanical degradation [16]. The DSC thermograms of the PP and WPC after nine recycling steps are shown in Figure 5 and a comparison of all materials is shown in Appendix A.

Both materials presented a slight decrease in the melting temperature as the number of recycling passes increased. The PP-0 registered a melt temperature around 171.5 °C while after nine extrusions recycles (PP-9) were around 167.3 °C, a decrease of 4.2 °C. Da Costa et al. [16] reported the same tendency, the reduction in the melting temperature for PP as the number of the extrusion cycles increased. The authors explained that the decrease in the melting temperature is because of the presence of small and degraded molecules on the recrystallized material. For the WPCs, the melting temperature between the WPC-0 and the WPC-9 sample decreased by only 1.6 °C. A similar trend has been reported for biocomposites, where the melting temperature was reduced with the increasing number of recycling passes [12]. The reduction in the melting temperature can be related to the decrease in the molecular mass due to the chain scission [22], in addition, the PP samples can exhibit a yellow color [16]. Due to the degradation of the wood, the WPC samples showed a darker color with an increase in the number of recycles [17]. The change in color for PP-0 and PP-9, as well as for WPC-0 and WPC-9, can be seen in Figure 6.

The TGA and derivative thermogravimetric (DTG) curves for PP and WPC before recycling (PP-0 and WPC-0), as well as for nine times recycling (PP-9 and WPC-9), are shown in Figure 7 and a comparison of all materials is shown in Appendix A. The PP samples exhibited only one stage of decomposition reaching a maximum peak at around 460 °C. Three stages of decomposition were found for all biocomposites. This started with dehydration at low temperatures, followed by the decomposition of hemicellulose starting at around 229 °C. The cellulose decomposition reached a maximum of 368 °C. The third stage was reached by the PP decomposition at around 468 °C.

Table 1 shows the onset degradation temperature and the maximum degradation peak (DTG_max_) temperatures for both materials. The increase in the number of recycles led to a slight decrease in the onset degradation temperature, corroborating with the literature [23]. The onset temperatures decreased by 1.4% for PP and 2.0% for biocomposites when comparing raw material with the ninth recycle sample. No significant differences in DTG_max_ temperature for both materials were observed. As can be seen, the variation in melting and degradation temperatures was not affected by the repeated extrusion process.

### 3.4. Mechanical Properties

The mechanical behaviors for PP and WPC are shown in Figure 8. The modulus is 78% higher (WPC-0 2.5 GPa compared to PP-0 1.4 GPa) and the strength is 64% better (46 MPa for WPC-0 compared with 28 MPa of the PP-0). These properties are excellent, especially the strength—the reason for the high reinforcing efficiency on this composite is because wood fibers are used instead of the wood flour which is more commonly used raw material in the WPC [9].

Regarding the effect of the repeating extrusions passes on the stiffness, both PP and WPC show similar behavior—a slight increase, but then a decrease to the initial level. The increase in the modulus is most likely due to the improved fiber dispersion. While the subsequent decrease could be due to the fiber breakage [17]. According to ANOVA and Tukey’s test, the changes are not statistically significant for the PP samples, however, for biocomposites, two pairs appear to be significantly different (WPC-0 and WPC-7 and WPC-1 and WPC-7), as shown in Table 2. During the recycling process, the enhancement of fiber dispersion and fiber breakage can occur, and these phenomena can influence the mechanical properties [12,13].

The data in Table 2 and Figure 8 show that repeated extrusions did not substantially affect the tensile strength of the PP. The most significant change was found for biocomposites, where the strength decreased from ≈ 46 MPa for WPC-0 to ≈ 37 MPa for WPC-9, a decrease of 20%. This reduction can be explained by the reduction of the fiber length and the aspect ratio, which consequently reduces the reinforcing efficiency [12]. According to the statistical analysis (Table 2), the two pairs for PP samples (PP-0 and PP-7 and PP-0 and PP-9) appear to be significantly different, confirming that besides the small change in the property, the recycling process has some effect on the strength. For WPCs, the statistical analysis showed a significant difference between all pairs of groups, validating the effect on the recycling process on the tensile strength.

For WPCs, the elongation had a small decrease when comparing WPC-0 with WPC-9, according to ANOVA and Tukey’s test, only one pair (WPC-3 and WPC-7) is statistically different. For PP samples, the elongation was found to have a minor increase as the number of recycles increased. Nonetheless, the statistical analysis showed a significant difference only between the pair PP-0 and PP-9. This increase could be because of the decreased molecular chain length, while in WPCs the slight decrease could be ascribed with improved adhesion to the wood fiber. The representative stress-strain curves for PP-0, PP-9, WPC-0, and WPC-9 are shown in Figure 9.

Overall, the mechanical properties of the starting WPC (WPC-0) are very good, when compared with earlier studies [2,12,24,25,26,27,28,29,30,31], as shown in Figure 10. Even after nine recycles, WPC-9 strength is higher than other WPCs found in the literature [24,25,27,30].

### 3.5. Interphase Interaction and Fracture Surfaces

Maleic anhydride as the coupling agent in the WPCs can interact with the free hydroxyl groups on the wood fiber surface to form new covalent bonds and hydrogen bonds across the fiber surface [32,33,34]. In an FTIR spectrum, OH stretching region can be found between 3000–3700 cm^−1^, equally, the hydrogen bonding interactions between MAPP and the –OH groups of wood fibers lead to a peak shift to lower wavenumber (3200–3500 cm^−1^) [32]. As shown in Figure 11, the IR absorption band of free –OH groups of wood fiber (TMP) was observed in 3352 cm^−1^. The spectrums shifted to 3340 cm^−1^ for WPC-0 and 3329 cm^−1^ for WPC-9, which confirms that more hydrogen bonds were generated during the recycling extrusion. The larger spectral shift in the spectrum of WPC-9 indicates that more hydrogen bonds were formed with the increase in the number of recycles [32]. The bands at 1375 and 1452 cm^−1^ are the characteristics of polypropylene [35]. In the case of covalent bonds, the reaction between the maleic anhydride and hydroxyl groups leads to ester bond formation [33,34]. The formed ester linkage can be identified by the generation of a new band around 1740 cm^−1^ [32,34]. However, due to the overlapping of the carbonyl C=O stretching of acetyl groups in hemicellulose and carbonyl aldehyde in lignin around 1738 cm^−1^ [36,37] that are components present in the TMP fibers, it is difficult to detect the band of the new ester linkage in the WPCs.

The fracture surfaces after the mechanical tests are shown in Figure 12. In the case of PP samples, the fracture surface of PP-9 (Figure 12b) was found to be more ductile than the PP-0 samples (Figure 12a). All fractured WPCs surfaces are shown in Appendix A. This can be attributed to the increase in the elongation after the recycling process, as observed in Table 2. The fracture surfaces of the initial WPC are shown in Figure 12c, where it is possible to see wood fibers and fiber pull-outs. While in Figure 12d, after nine extrusion passes, it is much more homogeneous and it is difficult to see any wood fibers, which is attributed to better dispersion, smaller fiber size, and improved interfacial adhesion (confirmed by IR spectra). The same tendency was reported by Shahi et al. [10], where the reduction in viscosity leads to a better distribution of the fibers in the melted polymer contributing to a more homogenous material. Hence, it is again confirmed from the SEM images that the main reason for the strength reduction is the decrease in the fiber aspect ratio, as discussed previously.

## 4. Conclusions

The changes in the mechanical, thermal, and rheological properties of polypropylene (PP) and its wood-polymer composites (WPCs), induced during the recycling process using an extruder, were analyzed and compared. The characterizations were carried out after one, three, five, seven, and nine recycles in order to investigate the degradation process.

In general, the melting temperature and the degradation temperature for both PP and WPC were hardly influenced by the recycling, indicating that these materials are suitable for repeated recycling using the extrusion process, which indicates longer service life of these WPCs. In addition, it was found that the recycling process affects the aspect ratio of the wood fiber, due to the shear forces occurring in the extruder. In the same way, shear and temperature contributed to the slight decrease in the molecular weight of the PP, as a result of the chain scission reactions. The tensile properties after recycling passes did not show great variations in the modulus for both materials. Admitting that the number of recycling passes has a detrimental effect on the WPC strength, which was noted as only a minor effect for PP samples, WPCs still presented not only better modulus but also higher strength values with respect to the unfilled PP, even after nine recycling passes. The fiber dispersion in WPC-9 was found to increase, due to the decrease in the fiber length but also due to the decrease in the polymer viscosity, enhancing the dispersion ability. Likewise, after nine reprocessing passes, the interface adhesion between wood fibers and polypropylene improved, which is confirmed by the FTIR spectroscopy.

Comparing the starting material (WPC-0) with other WPCs in the literature, the material presents very good mechanical properties, especially strength. After nine exposures in the extruder, the WPC-9 still can present higher strength than many previously reported WPCs. Overall, all results clearly demonstrate that the studied WPCs are highly recyclable without drastic changes in the properties. In addition, the same extrusion process used industrially for the manufacturing of polymers and WPCs can be used for the recycling of WPCs. This indicates that the WPCs consisting of TMP fibers and PP have great potential to contribute to sustainability.

## Figures and Tables

**Figure 1 polymers-12-01750-f001:**
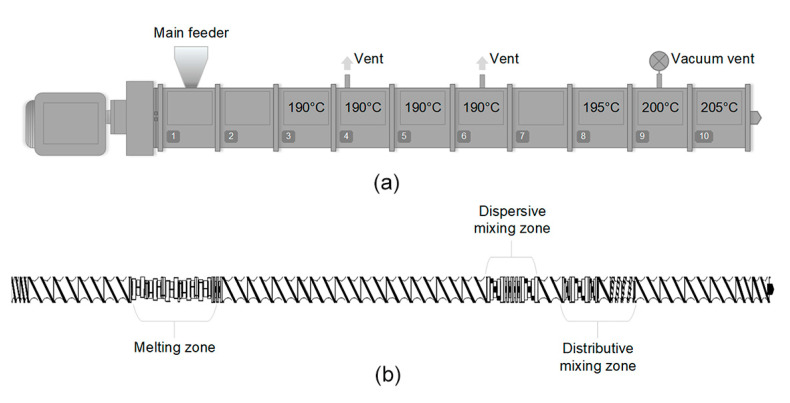
The used (**a**) twin-screw extruder profile and (**b**) screw-design in the recycling process.

**Figure 2 polymers-12-01750-f002:**
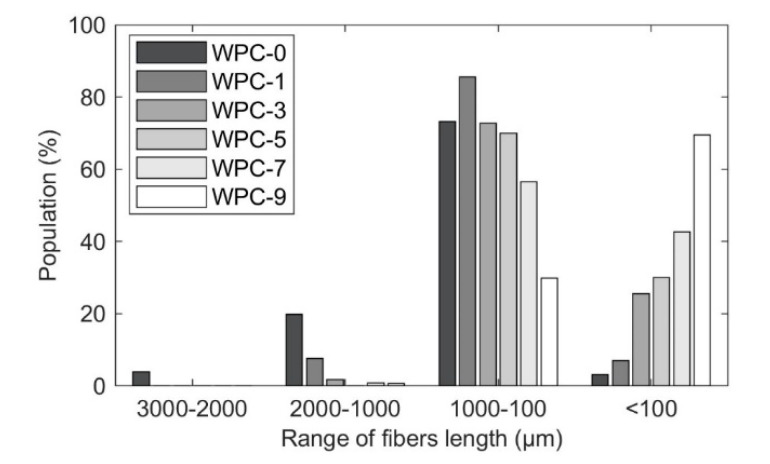
Fiber length distribution. Compacted-thermomechanical pulp (TMP) fibers extracted via Soxhlet from wood-fiber thermoplastic composites (WPCs).

**Figure 3 polymers-12-01750-f003:**
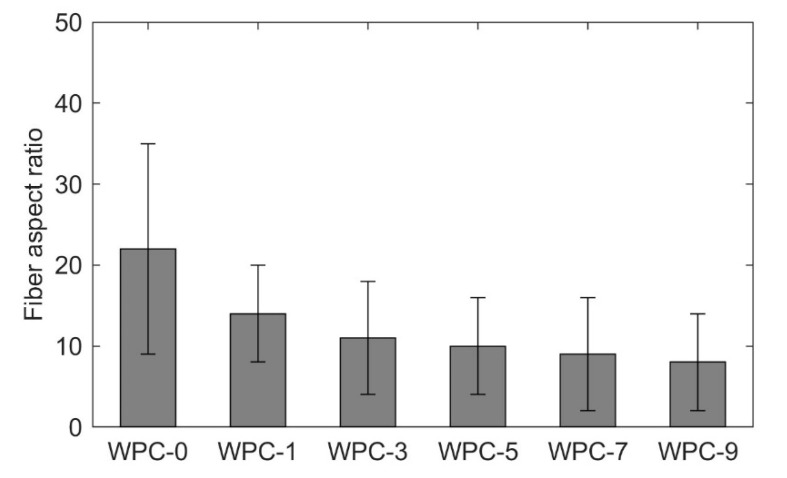
Fiber aspect ratio (compacted-TMP extracted from WPCs using Soxhlet).

**Figure 4 polymers-12-01750-f004:**
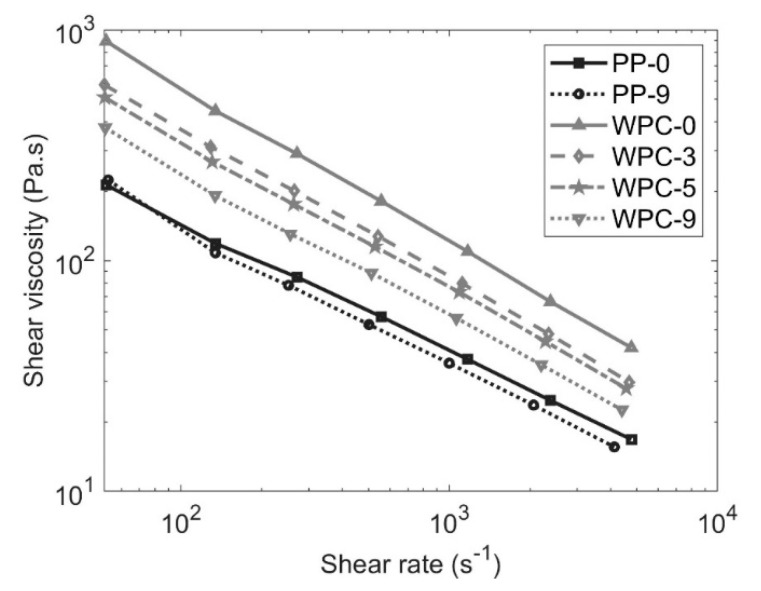
Shear viscosity curves for polypropylene (PP) and WPC before recycling and after recycling at 230 °C.

**Figure 5 polymers-12-01750-f005:**
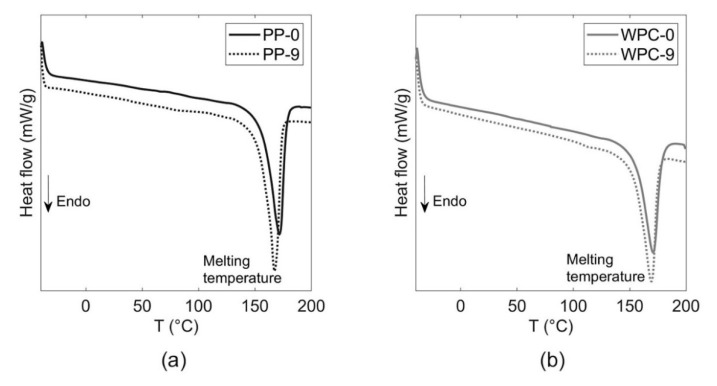
The effect of the heat flow—temperature graphs obtained from the tests performed with the differential scanning calorimetry (DSC) (**a**) PP and (**b**) WPC samples.

**Figure 6 polymers-12-01750-f006:**
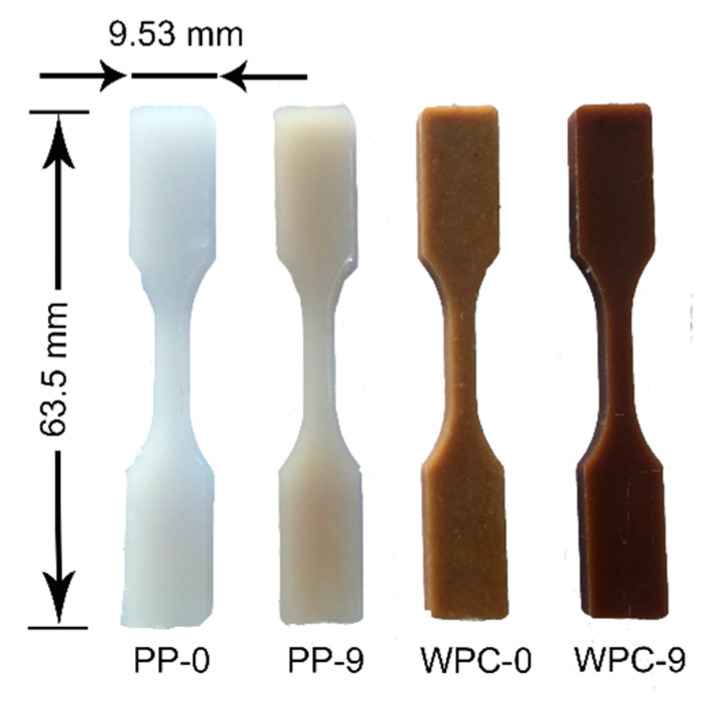
A color change in the PP and WPCs samples from the start through to nine recycling passes.

**Figure 7 polymers-12-01750-f007:**
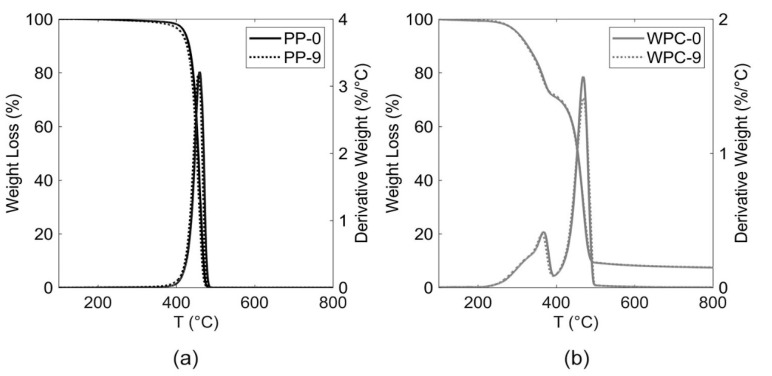
Thermo Gravimetric Analyzer (TGA)/derivative thermogravimetric (DTG) curves for (**a**) PP and (**b**) WPCs before and after nine recycles.

**Figure 8 polymers-12-01750-f008:**
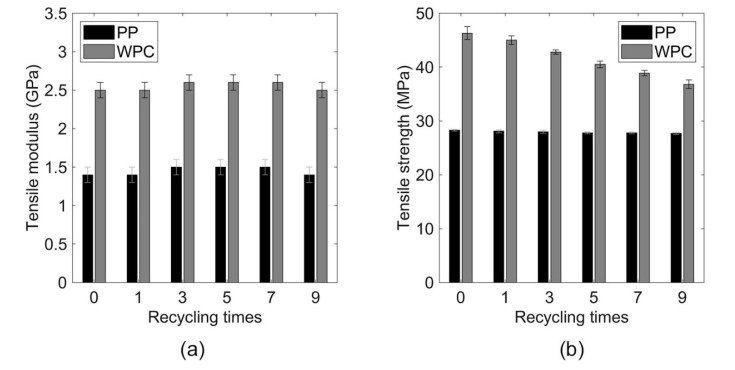
Tensile (**a**) and modulus (**b**) strength for PP and WPC samples before and after each recycling pass.

**Figure 9 polymers-12-01750-f009:**
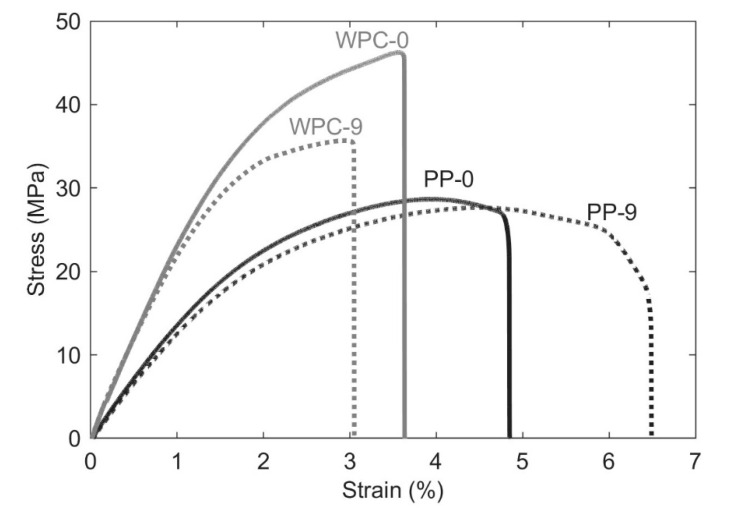
Stress-strain curves for PP-0, PP-9, WPC-0, and WPC-9 under tensile tests.

**Figure 10 polymers-12-01750-f010:**
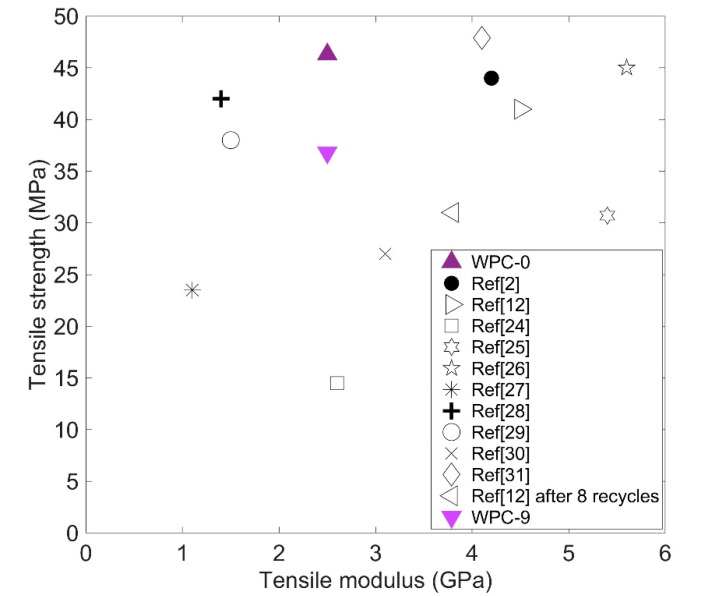
Comparative tensile strength versus modulus for different WPCs in the literature.

**Figure 11 polymers-12-01750-f011:**
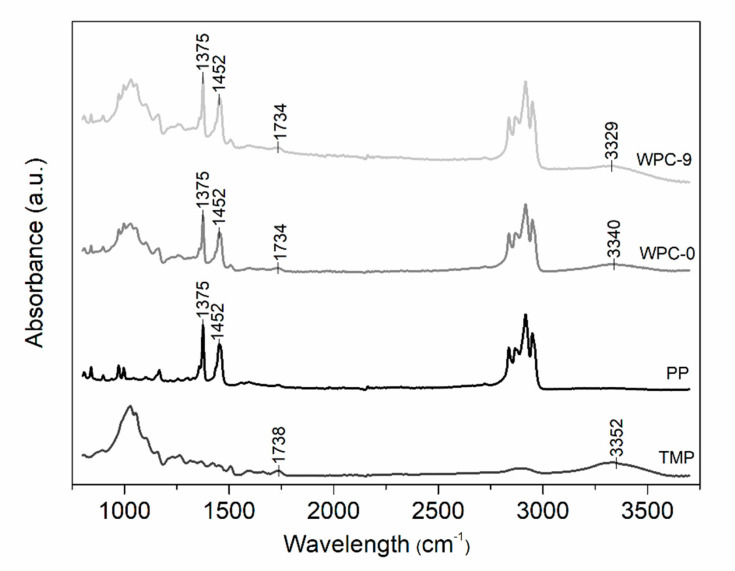
Fourier transform infrared (FTIR) spectra of TMP, PP, WPC-0, and WPC-9.

**Figure 12 polymers-12-01750-f012:**
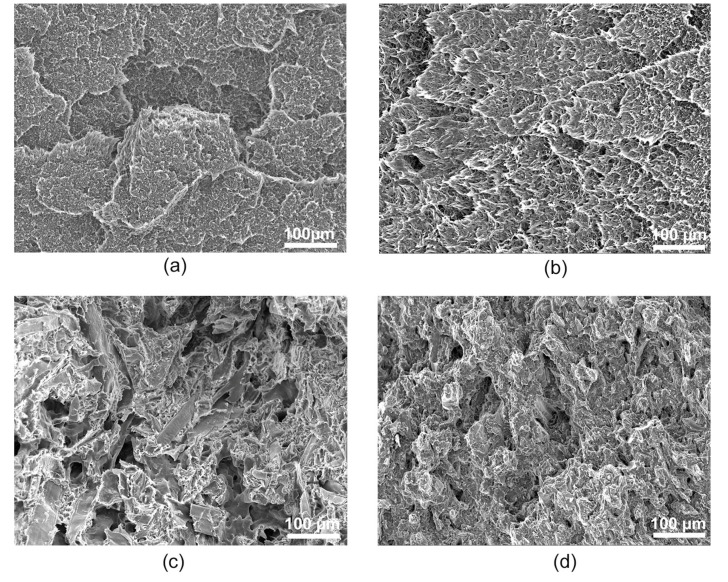
SEM images of fracture surfaces (**a**) PP-0, (**b**) PP-9, (**c**) WPC-0, and (**d**) WPC-9 (scale bar: 100 µm).

**Table 1 polymers-12-01750-t001:** Onset degradation temperatures (°C) and the maximum DTG peak temperature (°C) for PP and WPCs before and after recycling.

Recycling	T_onset_ (°C)		DTG_max_ (°C)	
	**PP**	**WPC**	**PP**	**WPC**
0	440	292	460	469
1	438	287	459	467
3	435	287	457	465
5	435	287	458	469
7	434	286	457	468
9	434	286	456	469

**Table 2 polymers-12-01750-t002:** Mechanical properties and statistical analysis of neat PP and WPC before and after recycling.

No. of Passes	Tensile Modulus (GPa)		Tensile Strength (MPa)		Elongation at Break (%)	
	PP	WPC	PP	WPC	PP	WPC
0	1.4 ± 0.1 ^A^	2.5 ± 0.1 ^A^	28.3 ± 0.2 ^A^	46.3 ± 1.2 ^A^	5.0 ± 1.3 ^A^	3.6 ± 0.7 ^A/B^
1	1.4 ± 0.1 ^A^	2.5 ± 0.1 ^A^	28.1 ± 0.3 ^A/B^	45.0 ± 0.8 ^B^	5.7 ± 1.0 ^A/B^	3.5 ± 0.2 ^A/B^
3	1.5 ± 0.1 ^A^	2.6 ± 0.1 ^A/B^	28.0 ± 0.3 ^A/B^	42.8 ± 0.4 ^C^	5.3 ± 0.5 ^A/B^	3.7 ± 0.4 ^A^
5	1.5 ± 0.1 ^A^	2.6 ± 0.1 ^A/B^	27.8 ± 0.2 ^A/B^	40.5 ± 0.6 ^D^	6.1 ± 0.6 ^A/B^	3.4 ± 0.3 ^A/B^
7	1.5 ± 0.1 ^A^	2.6 ± 0.1 ^B^	27.8 ± 0.2 ^B^	38.9 ± 0.5 ^E^	5.8 ± 0.5 ^A/B^	3.1 ± 0.1 ^B^
9	1.4 ± 0.1 ^A^	2.5 ± 0.1 ^A/B^	27.7 ± 0.2 ^B^	36.8 ± 0.8 ^F^	6.8 ± 0.6 ^B^	3.2 ± 0.4 ^A/B^

^A/B^ Marked with the same let er within the same column are not significantly different at 5% significant level based on ANOVA and Tukey’s test.

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
