# Peer review of "The Effect of Recycling on Wood-Fiber Thermoplastic Composites"

_polymers, 2020, doi:10.3390/polym12081750_

Round 1

Reviewer 1 Report

The work is interesting and results well presented. However, some the present data or figures are incomplete, I suggest some revisions before publication.
My questions and comments are shown as follow.

Q1.  In Figure 2, you should provide the shear rate dependence of the viscosity profile of WPC-1 and WPC-7.

.

Q2. In Figure 3, why not provide the DSC for all the WPC samples, including WPC-1 WPC-3, WPC-5 and WPC-7 ? 

Q3. In accordance with Table 1, all the TG/DTG curves for all the WPC samples, including WPC-1 WPC-3, WPC-5 and WPC-7, should be added.

  In addition, except for the onset degradation temperatures, I suggest to compare the peak values in DTG for all the samples about thermal degradation behavior.

Q4. Similarly, In Figure 11 and 12, why not provide the figure for all the WPC samples, including WPC-1 WPC-3, WPC-5 and WPC-7 ? I suggest to supplement.

Q5. In Figure 11, the unit of Absorbance should be added as “a.u.”

Author Response

see the document.

Reviewer 2 Report

The following issues need to be addressed before the ms. can be considered for publication:

  • L155. Please include standard deviations in Figure 2, as you do in Fig. 3, and indicate significant differences (after an ANOVA or a Kruskal-Wallis analysis, as appropriate).
  • L172. Figure 3. Indicate significant differences (if any) after an ANOVA or a Kruskal-Wallis analysis. Please comment on the large deviation obtained for WPC-0 in the text. Please note that a parenthesis is missing in the figure caption.
  • L263-L269 (including Fig. 9): the differences in behaviour between WCP0-WCP9 (decrease) and PPo-PP9 (increase) need to be explained and discussed in more detail.
  • Figure 11 does not add any useful information (the description provided in L277-L291 is enough). Please delete or move to supporting information.

Other minor issues:

  • Please correct the capitalization in subsection titles.

Author Response

see the response
